# The Antioxidant PAPLAL Protects against Allergic Contact Dermatitis in Experimental Models

**DOI:** 10.3390/antiox13060748

**Published:** 2024-06-20

**Authors:** Shuichi Shibuya, Kenji Watanabe, Takahiko Shimizu

**Affiliations:** Aging Stress Response Research Project Team, National Center for Geriatrics and Gerontology, 7-430 Morioka-cho, Obu 474-8511, Aichi, Japan; s-shibuya@ncgg.go.jp (S.S.); kng-wtnb@ncgg.go.jp (K.W.)

**Keywords:** antioxidants, oxidative stress, metal nanoparticles, allergic contact dermatitis, skin inflammation

## Abstract

PAPLAL, a mixture of platinum (nPt) and palladium (nPd) nanoparticles, is widely used as a topical agent because of its strong antioxidant activity. Allergic contact dermatitis (ACD) is one of the most common occupational skin diseases worldwide. However, the role of oxidative stress in ACD remains unclear. In the present study, we investigated the protective effects of topical PAPLAL treatment on 2,4-dinitrofluorobenzene (DNFB)-induced ACD. DNFB treatment increased 8-isoprostane content; upregulated *Xdh*, *Nox2*, and *Nox4*, pro-oxidant genes; and downregulated *Sod1*, an antioxidant gene, indicating oxidative damage in the ear skin. PAPLAL therapy significantly reduced ear thickness associated with the downregulation of inflammatory cytokine-related genes. PAPLAL also significantly increased the expression of the stress-response-related genes *Ahr* and *Nrf2,* as well as their target genes, but failed to alter the expression of redox-related genes. Furthermore, *Sod1* loss worsened ACD pathologies in the ear. These results strongly suggest that PAPLAL protects against ACD through its antioxidant activity and activation of the AHR and NRF2 axes. The antioxidant PAPLAL can be used as a novel topical therapy for ACD that targets oxidative stress.

## 1. Introduction

PAPLAL, a mixture of platinum (nPt) and palladium (nPd) nanoparticles, has been used to treat Japanese patients with various pathologies [1]. We previously reported that PAPLAL has both superoxide dismutase (SOD) and catalase activities and that topical application of PAPLAL improves skin atrophy caused by oxidative stress in an aging-mouse model [1,2]. The topical application of PAPLAL-containing cream also improved the disappearance of pigmentation in the skin of patients with vitiligo [3]. PAPLAL also suppressed H_2_O_2_-induced cellular senescence [4], indicating the protective action of catalase activity. Although Pd carries the risk of causing a metal allergy, PAPLAL, which contains nPd, does not induce a palladium allergy [5]. Thus, PAPLAL treatment is expected to have many beneficial effects in oxidative stress-mediated diseases.

Allergic contact dermatitis (ACD) is a common skin disease accompanied by cutaneous allergic signs and symptoms such as redness, scaling, dryness, edema, warmth, and pruritus [6,7,8]. ACD is a delayed-type hypersensitivity characterized by local inflammation of the skin triggered by exposure to irritants or low-molecular-weight allergens known as haptens [9]. ACD mainly involves two phases: sensitization and elicitation. The first reaction is the sensitization phase, followed by the elicitation phase, which is characterized by skin inflammation mediated by proinflammatory cytokines [10]. ACD is commonly treated with glucocorticoids, which also increase the risk of undesirable effects such as hypertension, diabetes, and cataracts [11]. Therefore, topical skin preparations that are safer and have a lower risk of side effects are expected to serve as therapeutic agents for ACD. Since the contribution of oxidative damage to ACD has remained largely unexplored, the therapeutic effect of antioxidants on ACD remains unclear.

In this study, we investigated the effect of PAPLAL on skin inflammation in ACD. Furthermore, we clarified the contribution of oxidative damage to ACD and evaluated the antioxidant and stress-response abilities of PAPLAL and ACD.

## 2. Materials and Methods

### 2.1. Reagents

PAPLAL was provided by Toyokose Pharmaceutical Co. (Tokyo, Japan) and Musashino Pharmaceutical Co. (Tokyo, Japan). PAPLAL is composed of a mixture of 0.2 mg/mL (1.03 mM) nPt and 0.3 mg/mL (2.82 mM) nPd. 1-Fluoro-2,4-dinitrobenzene (DNFB) was purchased from Merck (Darmstadt, Germany). PAPLAL solution was used without dilution (1×) or diluted to 1% with water (0.01×).

### 2.2. Mice

Female BALB/c mice were purchased from CLEA Japan (Tokyo, Japan). *Sod1*^−/−^ mice were purchased from Jackson Laboratory (Bar Harbor, ME, USA). Genotyping of the *Sod1*^−/−^ allele was performed via genomic polymerase chain reaction using genomic DNA isolated from the tail tip, as reported previously [12]. Female BALB/c and male *Sod1*^−/−^ mice were used at 6–10 weeks and 10–12 months old, respectively. The animals were housed under a 12 h light/dark cycle and fed ad libitum. Mice were maintained and studied according to protocols approved by the National Center for Geriatrics and Gerontology.

### 2.3. Induction of ACD and Treatment of PAPLAL

The abdominal skin of each mouse was shaved and sensitized with 50 μL DNFB in acetone/olive oil (4:1, *v*/*v*; FUJIFILM Wako Pure Chemical, Osaka, Japan) on days 1 and 2 (Figure 1). The mice were topically treated with 20 μL DNFB, which was applied to the surface of the left and right ears on day 6 (first challenge) and day 13 (second challenge). The concentration of DNFB administered to BALB/c mice was 0.5%. As *Sod1*^−/−^ mice were expected to have increased sensitivity to DNFB, the concentration of DNFB administered to *Sod1*^−/−^ mice was 0.3%. The vehicle group was treated with acetone/olive oil (4:1 *v*/*v*). Twenty milliliters of PAPLAL solution and water was applied to the ears daily from day 0 for the protective experiments. We measured ear thickness using a dial thickness gauge (Ozaki Mfg., Tokyo, Japan) daily during the first and second phases of ACD. A schematic illustration of ACD induction by DNFB and treatment with PAPLAL in BALB/c mice is shown in Figure 1.

### 2.4. 8-Isoprostane Contents

The ear tissue specimens were homogenized with 0.1 M phosphate (pH 7.4) containing 1 mM ethylenediaminetetraacetic acid (Dojindo Laboratories, Kumamoto, Japan) and 50 μg/mL (*w*/*w*) dibutylhydroxytoluene (FUJIFILM Wako Pure Chemical, Osaka, Japan). The homogenate was centrifuged at 8000× *g* for 10 min at 4 °C, and the total supernatant was used for the assay. The 8-isoprostane concentration of the homogenate was measured using an 8-isoprostane enzyme immunoassay kit (Cayman Chemical Company, Ann Arbor, MI, USA), according to the manufacturer’s instructions. The protein concentration of the supernatant was assessed using a DC-protein assay kit (BioRad, Hercules, CA, USA), and 8-isoprostane levels were normalized to the protein level.

### 2.5. Histological Analyses

For histological observations, ear tissue specimens obtained from mice were dissected, fixed in 20% formalin neutral buffer solution (FUJIFILM Wako Pure Chemical, Osaka, Japan) overnight, embedded in paraffin, and sectioned on a microtome at a thickness of 4 μm using standard techniques. Hematoxylin and eosin staining was performed as previously described [5]. The thickness of the ear specimens was measured using the ImageJ 1.53e software program (National Institutes of Health, Bethesda, MD, USA).

### 2.6. Quantitative Polymerase Chain Reaction (qPCR)

Total RNA was extracted from ear specimens using TRIzol reagent (Thermo Fisher Scientific, Waltham, MA, USA) according to the manufacturer’s instructions. cDNA was synthesized from 1 μg of total RNA using the ReverTra Ace qPCR RT Master Mix (Toyobo, Osaka, Japan). qPCR was performed using SsoAdvanced SYBR Green SuperMix (Bio-Rad, Hercules, CA, USA) on a CFC Ops 96 (Bio-Rad, Hercules, CA, USA), according to the manufacturer’s protocol. All data were normalized to the level of the housekeeping gene *Rps14*. When the PCR reaction produced multiple products or the PCR reaction efficiency was low, corresponding data were excluded from the analysis. The primer sequences used are listed in Table 1.

### 2.7. Treatment with PAPLAL in a Human Epidermal Skin Model

The human epidermal skin model (LabCyte EPI-MODEL12; J-TEC, Aichi, Japan) was cultured in accordance with the manufacturer’s instructions. The skin model was treated with 200 μL PAPLAL and incubated at 37 °C for 72 h. After incubation, the skin tissues and lower-conditioned medium were collected. The absorbance of the standard PAPLAL solution and the lower-conditioned medium was measured at 450 nm using a SpectraMax Paradigm (Molecular Devices, San Jose, CA, USA). The total number of nanoparticles in the lower-conditioned medium was calculated from the absorbance of the standard PAPLAL solution. The contents of nPt and nPd in the skin-model tissues and lower-conditioned medium were measured by inductively coupled plasma mass spectrometry (ICP-MS) using Agilent 8800 (Agilent Technologies Japan, Tokyo, Japan) at the UBE Scientific Analysis Laboratory, Inc. (Yamaguchi, Japan). The skin model tissue and plasma were thermally decomposed by adding sulfuric acid and nitric acid, and the residue was dissolved by adding nitric acid and hydrochloric acid. The obtained acid solution was diluted with ultrapure water and used for quantitative analyses.

### 2.8. Statistical Analyses

Statistical analyses were performed using Student’s *t*-test for comparisons between two groups, a one-way analysis of variance, and Tukey’s test for comparisons of three or more groups. Differences between the data were considered significant when *p*-values were less than 0.05. All data are expressed as mean ± standard deviation (SD).

## 3. Results

### 3.1. PAPLAL Improved Skin Inflammation in DNFB-Induced ACD

First, we investigated the protective effect of PAPLAL against skin inflammation using an ACD model induced by DNFB (Figure 1). Specifically, DNFB was applied to the ear to induce skin inflammation after sensitizing the mice with DNFB to the abdominal skin. In addition to DNFB sensitization and induction of contact dermatitis in the ears, we applied PAPLAL to the ears and measured changes in ear thickness. During the first challenge, DNFB treatment significantly increased ear thickness compared to that in the control groups, but additional treatment with PAPLAL failed to improve skin inflammation in the ears (Figure 2A). Furthermore, during the second challenge, DNFB treatment increased skin thickness and exacerbated the inflammatory response compared with the first challenge (Figure 2A). Additional application of PAPLAL at a high concentration in the second challenge inhibited hyperplasia of the ear skin, suggesting an inhibitory effect of PAPLAL on ACD (Figure 2A). In addition, the application of PAPLAL at a low concentration also inhibited the increase in skin thickness in the ears 24 h after the second challenge, showing a moderate protective effect on ACD (Figure 2A). Furthermore, we analyzed the content of 8-isoprostane, an oxidative stress marker, in ear tissues treated with or without DNFB and PAPLAL. Treatment with DNFB significantly upregulated the 8-isoprostane content in ear tissues (Figure 2B). Additional treatment with PAPLAL significantly decreased the 8-isoprostane content in ear tissues, indicating that PAPLAL attenuated the oxidative damage induced by DNFB (Figure 2B). Consistent with the histology of the ear tissue, PAPLAL significantly reduced ear thickness, as measured by quantitative measurements (Figure 2C).

Next, we examined whether or not PAPLAL treatment suppressed DNFB-induced inflammatory cytokines 72 h after the second challenge. Consistent with ear swelling, DNFB treatment increased the gene expression of *Il-6*, *Il-1β*, *Tnf-α*, *Il-17a*, and *Cxcl10* compared to the control (Figure 2D–H). Treatment with high-dose PAPLAL downregulated the levels of *Il-6*, *Tnf-α*, and *Cxcl10*, which had been upregulated by DNFB treatment (Figure 2D,F,H). In addition, administration of low-dose PAPLAL also significantly downregulated the gene expression of *Il-6*, *Il-1β*, *Tnf-α*, *Il-17a*, and *Cxcl10* (Figure 2D–H). These results indicate that PAPLAL protects against DNFB-induced skin inflammation.

### 3.2. PAPLAL Did Not Change Redox-Related Gene Expression in the ACD Model

To clarify the contribution of oxidative stress to DNFB-induced ACD, we analyzed the expression of genes associated with reactive oxygen species (ROS) metabolism in ear tissue. Although the expression of *Xdh*, the xanthine oxidoreductase gene, was significantly upregulated by DNFB treatment, PAPLAL failed to downregulate it (Figure 3A). The expression of NADPH oxidase (NOX) 2 and 4 was upregulated by DNFB treatment (Figure 3B,C). Additional treatment with PAPLAL failed to downregulate *Nox2* and *Nox4* expression (Figure 3B,C). Furthermore, we analyzed the expression of superoxide dismutase (SOD) 1, 2, and 3, which are the antioxidant enzymes. Although the *Sod1* gene was downregulated by DNFB treatment, *Sod2* showed significant upregulation (Figure 3D–E), suggesting different contributions to ACD among SOD families. Meanwhile, additional administration of PAPLAL induced no change in *Sod1*, *Sod2*, or *Sod3* gene expression (Figure 3D–F). Figure 2B shows that PAPLAL treatment suppressed 8-isoprostane accumulation in the ears of subjects with ACD. Taken together, these results indicate that PAPLAL improves ACD by reducing oxidative stress.

### 3.3. PAPLAL Increased the Expression of the Stress Response-Related Genes Ahr and Nrf2

A previous report showed that PAPLAL activates aryl hydrocarbon receptor (AHR) and nuclear factor erythroid 2-related factor 2 (NRF2), which are stress-response-related genes, in cultured keratinocytes [13]. We investigated the contribution of AHR and NRF2 pathways in an ACD mouse model. DNFB treatment did not alter the *Ahr* gene expression in ear skin (Figure 3G). In contrast to *Ahr*, DNFB treatment significantly downregulated AHR-targeting *Cyp1a1* expression (Figure 3H). Treatment with high concentrations of PAPLAL significantly increased the expression of both *Ahr* and *Cyp1a1* in ACD-induced ear tissues (Figure 3G,H). Furthermore, the gene expression of both *Nrf2* and its downstream target *Nqo1* were significantly increased by DNFB treatment (Figure 3I,J). PAPLAL treatment of ACD-induced ears significantly upregulated both the *Nrf2* and *Nqo1* gene expression (Figure 3I,J). These results indicate that, in addition to antioxidant activity, PAPLAL may also contribute to the improvement of skin inflammation in ACD by stress-response genes through AHR and NRF2 transcriptional factors.

### 3.4. Sod1 Deficiency Exacerbated DNFB-Induced ACD

In the present study, we observed that DNFB treatment suppressed *Sod1* expression (Figure 3D). We also demonstrated that *Sod1*-deficient mice exhibit various age-related pathologies, such as skin atrophy due to the accumulation of ROS [2,12,14,15]. We predicted that the significant accumulation of ROS due to *Sod1* loss would exacerbate skin inflammation in contact dermatitis. To further investigate the relationship between ROS and skin inflammation in ACD, we treated *Sod1*-deficient mice with DNFB and evaluated skin inflammation (Figure 4A). Notably, *Sod1*^−/−^ mice treated with DNFB exhibited markedly thickened ear skin compared to *Sod1*^+/+^ mice treated with DNFB in both the first and second challenges (Figure 4B,C), demonstrating exacerbation of inflammation by *Sod1* loss.

### 3.5. nPt and nPd Contained in PAPLAL Penetrated the Epidermal Layer in a Human Epidermal Model

Next, to investigate whether or not the nPt and nPd contained in PAPLAL permeate into the skin, we applied PAPLAL to the human skin epidermal model (Figure 5A). Because PAPLAL is a black color solution (Figure 5A), we first measured the absorbance (450 nm) of the medium. At 24 h after PAPLAL treatment, no nanoparticles were detected in the conditioned medium (Figure 5B). Meanwhile, approximately 1.48 μg of nanoparticles were detected in the lower medium after 72 h PAPLAL, indicating time-dependent skin permeability of PAPLAL (Figure 5B). To further clarify the skin permeability of nPt and nPd, we quantified the amounts of nPt and nPd in the epidermal tissue and lower medium by ICP-MS 72 h after PAPLAL treatment. The epidermal tissue treated with PAPLAL for 72 h contained approximately 2.70 μg and 8.33 μg of nPt and nPd, respectively (Table 2). Furthermore, the lower medium after 72 h of PAPLAL treatment contained approximately 0.04 μg and 0.35 μg of nPt and nPd, respectively (Table 2). The amounts of nPt and nPd that migrated into the lower medium were 0.28% and 0.59% of the amount contained in the PAPLAL, respectively (Table 2). These results suggest that the nPt and nPd contained in PAPLAL penetrate the epidermis and reach the dermal layer.

## 4. Discussion

In the present study, PAPLAL significantly reduced skin inflammation in DNFB-induced ACD (Figure 2) but failed to change the expression of redox-related genes compared to vehicle mice (Figure 3A–F), indicating that direct ROS-scavenging activity by PAPLAL contributes to the attenuation of inflammation in ACD. Accumulating evidence suggests that the administration of astaxanthin and N-acetylcysteine, which have antioxidant activities, also restores ACD induced by DNFB or trinitrochlorobenzene [16,17]. In contrast to antioxidants, oxidized dietary oils enhance DNFB-induced ear inflammation [18]. Thus, antioxidants such as PAPLAL can be applied to the skin and may have applications as novel anti-inflammatory therapeutics. Oxidative stress is an important factor involved in skin inflammation [19]. In an atopic dermatitis model treated with a mite antigen, ROS accumulates at the dermatitis site [20]. Furthermore, IL-13-/IL-4-induced dermatitis produces ROS through the activation of dual oxidase protein 1 (DUOX1) [21]. The thiol-reactive sensitizer DNFB, used as an ACD inducer, directly reacts with cytoplasmic glutathione (GSH), causing its rapid and marked depletion, which results in a general increase in ROS accumulation [22]. In the present study, we demonstrated that the accumulation of oxidative stress via the alteration of multiple redox genes also contributes to the development of ACD induced by DNFB. In particular, *Sod1* expression was significantly decreased by DNFB treatment (Figure 3D), indicating that DNFB shifts the skin to an oxidative state. In fact, we found that DNFB treatment increased 8-isoprostane content in the skin, and PAPLAL treatment significantly attenuated 8-isoprostane content in ACD models, although low-dose PAPLAL treatment had no such effect (Figure 2B). We found that 8-isoprostane content in the skin showed wide deviation compared with that of plasma (unpublished results). Previously, we reported that *Sod1*-deficient skin showed increased accumulation of 8-isoprostane [2,23]. *Sod1*-deficient mice exhibited exacerbated skin thickening in DNFB-induced ACD (Figure 4B,C), indicating the contribution of oxidative damage to ACD. We also reported that PAPLAL fully rescued aging-like skin atrophy in *Sod1*-deficient mice by reducing oxidative stress [1,2]. In this context, we speculate on the improvement of ACD pathologies by PAPLAL treatment in *Sod1*^−/−^ mice.

NRF2 is a transcriptional factor activated by oxidative stress that regulates oxidative stress-adaptive responses by upregulating the expression of antioxidant enzymes such as glutathione. In the present study, *Nrf2* expression was significantly increased by DNFB treatment (Figure 3I), suggesting a response to increased oxidative stress in ACD. ACD induction by 2,4-dinitrochlorobenzene treatment in *Nrf2*-deficient mice decreased antioxidant-related genes, increased chemokine production, and exacerbated skin inflammation with increased neutrophil recruitment, indicating a protective effect of NRF2 against ACD [24]. Notably, PAPLAL further increased *Nrf2* gene expression, which was upregulated in ACD (Figure 3I). Plant phenols, such as flavanols, which have antioxidant activity, activate NRF2, indicating a link between antioxidants and NRF2 activity [25]. PAPLAL contributes to ACD protection through additive redox regulation via direct antioxidant activity and NRF2 activation.

PAPLAL also upregulated the *Ahr* expression in ACD-induced ear tissues (Figure 3G,I). Our results are consistent with previous reports that PAPLAL upregulates AHR and NRF2 expression in cultured keratinocytes [13]. AHR, a ligand-activated transcriptional factor widely expressed at barrier sites and in the immune system, contributes to the maintenance of barrier function and homeostasis in the lungs and intestines [26,27]. AHR regulates the expression of skin barrier-related proteins, such as filaggrin, loricrin, and involucrin [28,29,30,31], and also plays an important role in skin barrier integrity and immunity [32,33,34,35,36,37,38,39]. AHR knockout mice show a loss of transepidermal water and aberrant expression of genes related to barrier integrity in keratinocytes, suggesting that AHR plays an important role in maintaining the homeostasis of skin function [33]. In atopic dermatitis, the AHR axis and IL-13/IL-4-JAK-STAT6/STAT3 axis compete to regulate the expression of skin barrier-related proteins [28,37,38,39]. The induction of significant inflammatory responses in ACD may be an aberrant immune response mediated by the altered expression of AHR target genes. Induction of ACD by DNFB treatment downregulated the expression of only the *Cyp1a1* gene and not the *Ahr* gene (Figure 3G,I). Because these gene expression analyses were performed during the last phase after ACD induction, gene expression during the acute phase must also be analyzed. Natural AHR ligands are largely derived from dietary indoles and tryptophan metabolism [40]. Coal tar and soybean tar, which have been used to treat dermatitis, are AHR ligands that enhance filaggrin expression [37,39]. Oral administration of 2-(1′H-indole-3′-carbonyl)-thiazole-4-carboxylic acid methyl ester, an AHR ligand, to DNFB-induced ACD mouse models improves skin inflammation by suppressing thymic stromal lymphopoietin (*Tsle*) gene expression via ARH activation [41]. PAPLAL not only increases the expression of *Ahr* but also acts as a ligand.

We also demonstrated that the nPt and nPd contained in PAPLAL permeated the cultured human epidermal model after 72 h of culture (Figure 5B). This result suggests that nPt and nPd administered to the skin penetrated the epidermis and reached the dermal layer. The particle sizes of nPt and nPd contained in PAPLAL are 1.93 ± 0.34 and 3.59 ± 0.56 nm, respectively [42]. Nanoparticles can pass between cells in the skin at the nano-level [43]. In the human epidermal model treated with PAPLAL for 24 h, no epidermal permeation of nPt and nPd was observed (Figure 5B). Furthermore, PAPLAL failed to reduce ear thickness during the first challenge with DNFB-induced ACD (Figure 2A). These results indicate that ear tissue has a lower permeability to nanoparticles than the cultured epidermal model and requires more continuous administration of PAPLAL. Furthermore, the permeability of nanoparticles may be increased in the skin, where the barrier function is impaired in the second phase of ACD. The nPt and nPd contained in PAPLAL are expected to reach the dermis and act directly on skin cells. Since PAPLAL is a noble metal with high-molecular-weight nanoparticles (particle size: about 2–4 nm [42]) compared to other antioxidants, including vitamin C, E, and NAC, its properties may influence biological and physiological dynamics in a unique manner in vivo. The protective effect of PAPLAL against ACD requires comprehensive consideration of the ability of the nanoparticles to penetrate the dermis and their subsequent bioavailability.

Chemical compounds such as DNFB, oxazolone, and picryl chloride have been used to induce ACD [44,45,46]. Recently, it was demonstrated that the mechanism of ACD differs depending on the substance acting as an allergen. DNFB used in the present study directly activates mast cells, resulting in the release of physiologically active substances with pro-inflammatory effects, including histamine [47]. Unfortunately, we could not evaluate the immune cell activity when PAPLAL was used as a model of DNFB-induced ACD. Further detailed analyses are needed to investigate the relationship between mast cell activation and oxidative stress.

## 5. Conclusions

In the present study, we demonstrated the important contribution of oxidative stress in ACD. Topically applicable antioxidants can be used as new therapeutic agents for ACD to replace steroids. PAPLAL has multiple protective effects against ACD, including direct antioxidant activity and the regulation of stress-response genes. Since the nanoparticles contained in PAPLAL can remain stable for a long time with strong antioxidant activity, PAPLAL is expected to be applied to new ACD treatment strategies targeting oxidative stress.

## Figures and Tables

**Figure 1 antioxidants-13-00748-f001:**
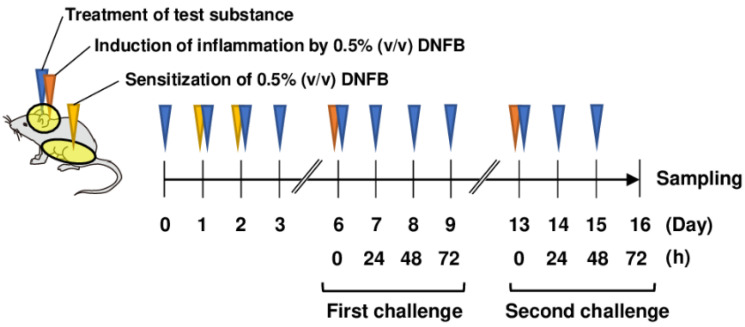
Treatment of PAPLAL in DNFB-induced ACD model mice. Schematic illustration of PAPLAL treatment in a DNFB-induced ACD mouse model. The abdominal skin of each mouse was shaved and sensitized with 50 μL DNFB in acetone/olive oil (4:1, *v*/*v*) on days 1 and 2. The mice were topically treated with 20 μL DNFB, which was applied to the surface of the left and right ears on day 6 (first challenge) and day 13 (second challenge). The vehicle group was treated with acetone/olive oil (4:1 *v*/*v*). Twenty milliliters of PAPLAL solution or water was applied to the ears daily from day 0 for the protective experiments.

**Figure 2 antioxidants-13-00748-f002:**
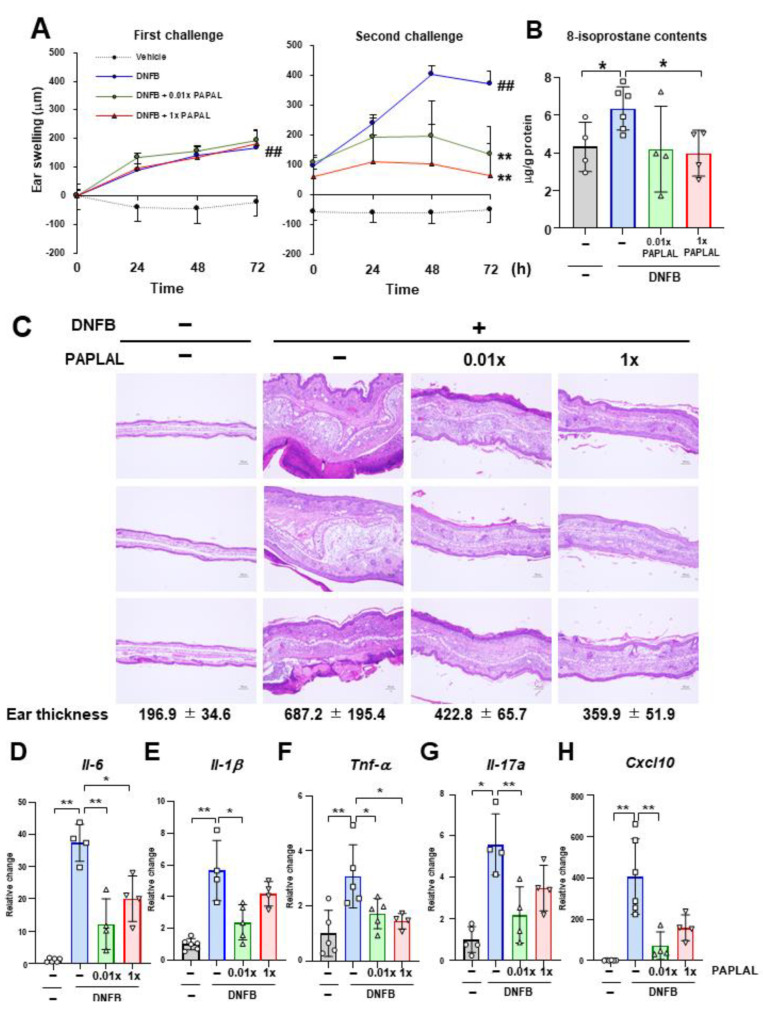
PAPLAL suppresses skin inflammation induced by DNFB. (**A**) Ear thickness of mice treated with or without PAPLAL in first (left) and second (right) challenges of ACD induced by DNFB. ^##^
*p* < 0.01 vs. vehicle. ** *p* < 0.01 vs. DNFB. Data are shown as the mean ± SD. (**B**) 8-isoprostane content of the ear tissue at day 16. (**C**) Ear sections were stained with hematoxylin and eosin on day 16, and quantified ear thickness histologically. The scale bars represent 100 μm. Relative change in the gene expression of *Il-6*. (**D**), *Il-1β* (**E**), *Tnf-α* (**F**), *Il-17a* (**G**), and *Cxcl10* (**H**) in ear tissue. * *p* < 0.05, ** *p* < 0.01. Data are shown as the mean ± SD (*n* = 4–6).

**Figure 3 antioxidants-13-00748-f003:**
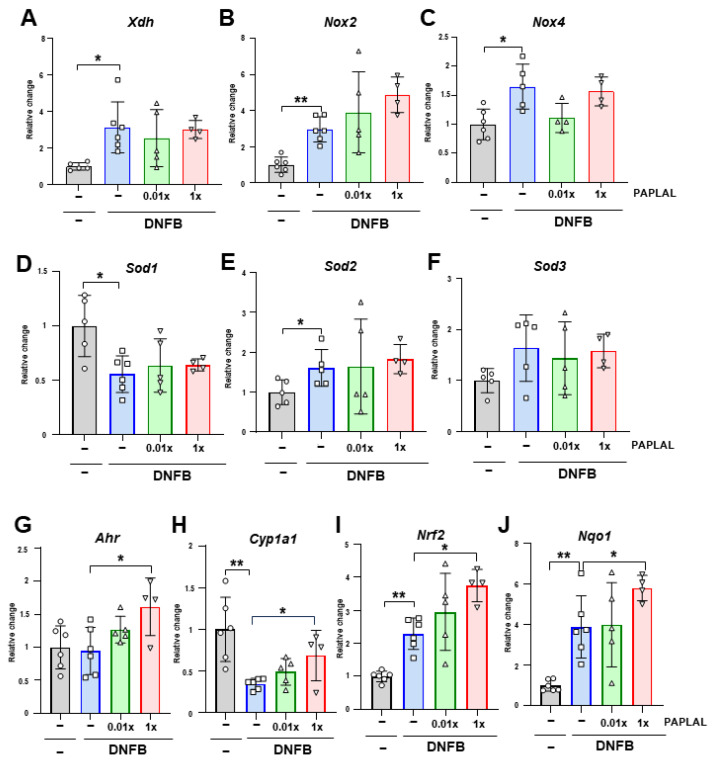
Expression analyses of redox and stress-response-related genes. Relative changes in the gene expression of *Xdh* (**A**), *Nox2* (**B**), *Nox4* (**C**), *Sod1* (**D**), *Sod2* (**E**), *Sod3* (**F**), *Ahr* (**G**), *Cyp1a1* (**H**), *Nrf2* (**I**), and *Nqo1* (**J**) in ear tissue. * *p* < 0.05. ** *p* < 0.01. Data are presented as the mean ± SD (*n* = 4–6).

**Figure 4 antioxidants-13-00748-f004:**
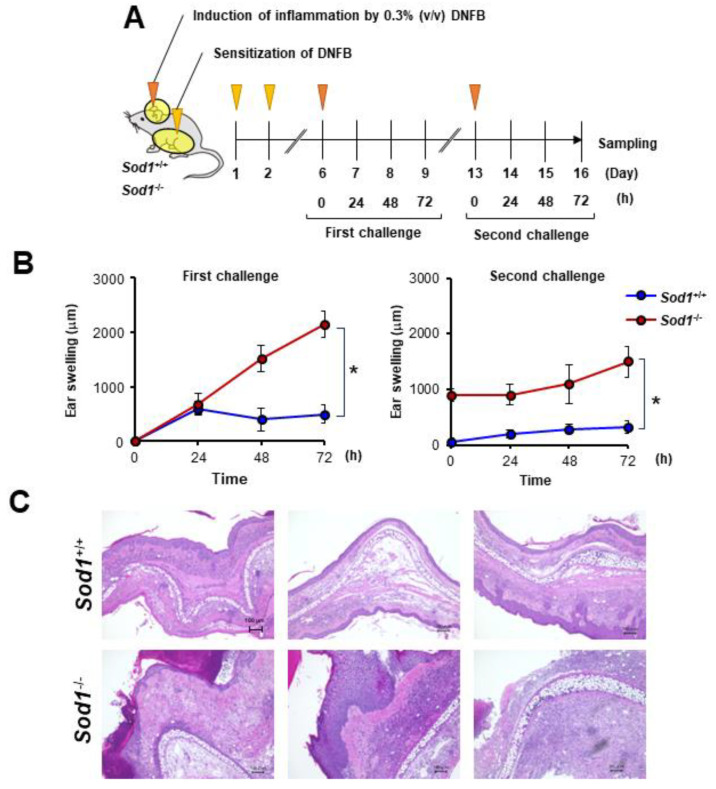
*Sod1* deficiency deteriorates skin inflammation induced by DNFB. (**A**) Schematic illustration of the induction of ACD by DNFB in *Sod1*^+/+^ and *Sod1*^−/−^ mice. Abdominal skin of *Sod1*^+/+^ and *Sod1*^−/−^ mice was shaved and sensitized with 50 μL DNFB in acetone/olive oil (4:1, *v*/*v*) for days 1 and 2. The mice were topically treated with 20 μL DNFB, which was applied to the surface of the left and right ears on day 6 (first challenge) and day 13 (second challenge). The vehicle group was treated with acetone/olive oil (4:1, *v*/*v*). (**B**) Ear thickness of *Sod1*^+/+^ and *Sod1*^−/−^ mice in the first (left) and second (right) challenge of ACD induced by DNFB. (**C**) Ear sections were stained with hematoxylin and eosin on day 16. The scale bars represent 100 μm. * *p* < 0.05. Data are presented as the mean ± SD.

**Figure 5 antioxidants-13-00748-f005:**
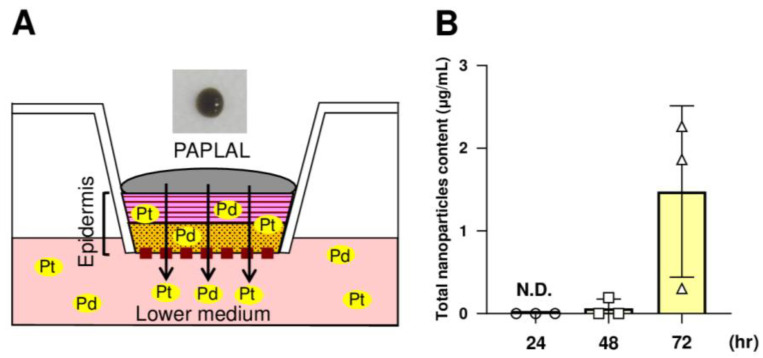
The nPt and nPd contained in PAPLAL penetrate the epidermis. (**A**) A human epidermal skin model was used in ex vivo experiments. (**B**) Quantification of the total nanoparticles content penetrating the lower-conditioned medium of the human epidermal model treated with PAPLAL by absorbance (450 nm). The total number of nanoparticles contained in the lower-conditioned medium was calculated from the absorbance of the standard PAPLAL solution. Data are presented as the mean ± SD.

**Table 1 antioxidants-13-00748-t001:** qPCR primers.

Gene Name	Accession Number	Forward	Reverse
*Ahr*	NM_013464	CTGGTTGTCACAGCAGATGCCT	CGGTCTTCTGTATGGATGAGCTC
*Cxcl10*	NM_021274	CACCATGAACCCAAGTGCTG	GGATAGGCTCGCAGGGATGA
*Cyp1a1*	NM_001136059	CATCACAGACAGCCTCATTGAGC	CTCCACGAGATAGCAGTTGTGAC
*Il-1β*	NM_008361	ATGGCAACTGTTCCTGAACTCAACT	CAGGACAGGTATAGATTCTTTCCTTT
*Il-6*	NM_031168	GCTACCAAACGTGATATAATCAGGA	CCAGGTAGCTATGGTACTCCAGAA
*Il-17a*	NM_010552	CAGACTACCTCAACCGTTCCAC	TCCAGCTTTCCCTCCGCATTGA
*Nox2*	NM_007807	TGGCGATCTCAGCAAAAGGTGG	GTACTGTCCCACCTCCATCTTG
*Nox4*	NM_015760	CGGGATTTGCTACTGCCTCCAT	GTGACTCCTCAAATGGGCTTCC
*Nrf2*	NM_010902	TTTTCCATTCCCGAATTACAGT	AGGAGATCGATGAGTAAAAATGGT
*Nqo1*	NM_008706	GCCGAACACAAGAAGCTGGAAG	GGCAAATCCTGCTACGAGCACT
*Rps14*	NM_020600	AGGAGTCTGGAGACGACGAT	CAGTCACTCGGCAGATGGTT
*Sod1*	NM_011434	AACCATCCACTTCGAGCAGAA	GCTGGCCTTCAGTTAATCCTGTA
*Sod2*	NM_013671	CTGGACAAACCTGAGCCCTAAG	AAGACCCAA AGTCACGCTTGA
*Sod3*	NM_011435	CTCTTGGGAGAGCCTGACA	GCCAGTAGCAAGCCGTAGAA
*Tnf-α*	NM_013693	ATGAGCACAGAAAGCATGATCCGC	GCTTGGTGGTTTGCTACGAC
*Xdh*	NW_011723	GCTCTTCGTGAGCACACAGAAC	CCACCCATTCTTTTCACTCGGAC

**Table 2 antioxidants-13-00748-t002:** Penetration of Pt and Pd nanoparticles in epidermal skin model.

	nPt		nPd	
Migration	(μg)	(%)	(μg)	(%)
Epidermal tissue	2.70 ± 0.56	6.77 ± 1.42	8.33 ± 0.58	11.87 ± 2.48
Lower medium	0.04 ± 0.06	0.28 ± 0.16	0.35 ± 0.13	0.59 ± 0.23

The human epidermal skin model was treated with 200 μL of PAPLAL and incubated at 37 °C for 72 h. After incubation, the skin model tissues and lower-conditioned medium were collected. The contents of nPt and nPd in the skin model tissues and lower-conditioned medium were measured by ICP-MS.

## Data Availability

Data are contained within the article.

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
