# Peer review of "The Antioxidant PAPLAL Protects against Allergic Contact Dermatitis in Experimental Models"

_antioxidants, 2024, doi:10.3390/antiox13060748_

Round 1
Reviewer 1 Report
Review report to The antioxidant PAPLAL protects against allergic contact dermatitis in mice by Shuichi Shibuya, Kenji Watanabe and Takahiko Shimizu:
In the presented study, the effects of a platinum and palladium nanoparticle mixture (PAPLAL) were investigated for application as antioxidant in allergic contact dermatitis. Further the antiinflammatory effect of PAPLAL was investigated. ACD was induced by DNFB in a mouse model and increased the 8-isoprostane content as a marker of oxidative stress. Concomitant application of PAPLAL resulted in reduced ear swelling and downregulation of inflammatory cytokine expression while stress response genes (AHR, NRF2) were upregulated. The role of oxidative stress in the ACD development was further underlined by using SOD knock out mice. The skin penetration of PAPLAL was shown by a human epidermal perfusion model.
The manuscript is written in a comprehensive and concise manner. The figures present the findings clearly and in a uniform colour scheme. The discussion reflects on the findings and related research. I only have a few remarks.
In this study the authors introduced the SOD-/- mice and showed an impressive ear swelling response when treated with DNFB. This was motivated by the observed decrease in expression of SOD1 after DNFB challenge. To strengthen the statement that PAPLAL shows antioxidative activity it would be interesting to see how PAPLAL application could compensate the SOD knock out. May be some discussion to it could be added.
Why did the authors use different concentrations of DNFB for challenging ACD (0.5% and 0.3% in the two experimental settings?
Content of 8-isoprostane was used as a marker for oxidative stress in the tissue. The analysis (Fig2B) revealed no difference in its content with different concentrations of PAPLAL. Could the author please discuss this finding and propose some ideas why this is the case.
line 119, chapter 2.7: „To show the penetration of nPT and nPd into and through the skin lower conditioned medium were measured by ICP-MS using Agilent 8800..“à Please name the method. It might not be familiar to all readers.
Author Response
Reviewer 1
The manuscript is written in a comprehensive and concise manner. The figures present the findings clearly and in a uniform colour scheme. The discussion reflects on the findings and related research. I only have a few remarks.
In this study the authors introduced the SOD-/- mice and showed an impressive ear swelling response when treated with DNFB. This was motivated by the observed decrease in expression of SOD1 after DNFB challenge. To strengthen the statement that PAPLAL shows antioxidative activity it would be interesting to see how PAPLAL application could compensate the SOD knock out. May be some discussion to it could be added.
Response: We previously reported that PAPLAL improves oxidative stress-induced skin atrophy in SOD1-deficient mice. Therefore, we expected that PAPLAL would also ameliorate ACD in SOD1-deficient mice. We have now added a description of the contribution of PAPLAL to ACD in SOD1-deficient mice to the ‘Discussion’.
(Lines 277-279 in Discussion) ‘We also reported that PAPLAL fully rescued aging-like skin atrophy in Sod1-deficient mice by reducing oxidative stress [1,2]. In this context, we speculate the improvement of ACD pathologies by PAPLAL treatment in Sod1-/- mice.’
Why did the authors use different concentrations of DNFB for challenging ACD (0.5% and 0.3% in the two experimental settings?
Response: We predicted that SOD1-deficient mice, which frequently develop skin inflammation, would have increased reactivity to DNFB application, so we reduced the DNFB concentration to 0.3%. As suggested, we have now added detailed descriptions of the DNFB concentration to the ‘Materials and Methods’.
(Lines 71-73 in Materials and Methods) ‘The concentration of DNFB administered to BALB/c mice was 0.5%. As Sod1-/- mice were expected to have increased sensitivity to DNFB, the concentration of DNFB administered to Sod1-/- mice was 0.3%.’
Content of 8-isoprostane was used as a marker for oxidative stress in the tissue. The analysis (Fig2B) revealed no difference in its content with different concentrations of PAPLAL. Could the author please discuss this finding and propose some ideas why this is the case.
Response: In the present study, administration of low-dose of PAPLAL tended to reduce the amount of 8-isoprostane, but the difference was not statistically significant. In our biochemical analyses, 8-isoprostane content in the skin showed wide deviation compared with that of plasma (unpublished results). We have now added a description of the 8-isoprostene levels in the skin during biochemical analyses to the ‘Discussion’.
(Lines 270-274 in Discussion) ‘In fact, we found that DNFB treatment increased 8-isoprostane content in skin, and PAPLAL treatment significantly attenuated 8-isoprostane content in ACD models, although low-dose PAPLAL treatment had no such effect (Figure 2B). We have that 8-isoprostane content in skin showed wide deviation compared with that of plasma (unpublished results)’
line 119, chapter 2.7: „To show the penetration of nPT and nPd into and through the skin lower conditioned medium were measured by ICP-MS using Agilent 8800..“à Please name the method. It might not be familiar to all readers.
Response: As suggested, we have added the full name of the ICP-MS to the ‘Materials and Methods’.
Reviewer 2 Report
This study revealed the “The antioxidant PAPLAL protects against allergic contact dermatitis in mice.”. The study will be beneficial for the literature. The report is an interesting study, but it needed some suggestions for publication.
1)In the method section: In the description of the statistics, the p value is less than 0.05, but in the figure captions it is different... what does it mean Data are presented as the mean ± SD (n = 4-6) - line188; when is it 4 and when is it 6? 2) Add the limitations of the study. 3)The conclusion section must be added to report critical comments on the results. A few references are from a time long past.
Author Response
Reviewer 2
This study revealed the “The antioxidant PAPLAL protects against allergic contact dermatitis in mice.”. The study will be beneficial for the literature. The report is an interesting study, but it needed some suggestions for publication.
Looking at the title, the work should include studies on mice, but the work also contains data on PAPLAL treatment in a human epidermis model, which should be included in the title or at least in the keywords.
Response: As suggested, we have changed "in mice" in the title to "in models".
- -1 In the method section: In the description of the statistics, the p value is less than 0.05, but in the figure captions it is different... what does it mean.
Response: We apologize for the confusion. In the present study, differences between the data were considered significant when P-values were <0.05. As suggested, we have removed the P values greater than 0.05 from Figures 2 and 3. Furthermore, we have removed the description concerning significant differences in the Sod3 gene from the ‘Results (Lines 180-182)’.
-2 Data are presented as the mean ± SD (n = 4-6) - line188; when is it 4 and when is it 6?
Response: In the present study, we used ACD models using the following numbers of BALB/c mice: Vehicle; 6, DNFB; 6, DNFB+0.01x PAPLAL; 5, DNFB+1x PAPLAL; 4. When PCR reaction produced multiple products or the PCR reaction efficiency was low, corresponding data were excluded from the analysis. We have now mentioned the selection criteria for RT-PCR data in the ‘Materials and Methods’.
(Lines 110-112 in Materials and Methods) ‘When PCR reaction produced multiple products or the PCR reaction efficiency was low, corresponding data were excluded from the analysis.’
- Add the limitations of the study.
Response: As suggested, we have added the limitations of the study to the ‘Discussion’.
(Lines 328-333 in Discussion) ‘Since PAPLAL is a noble metal with high-molecular-weight nanoparticles (particle size: about 2-4 nm [42]) compared to other antioxidants, including vitamin C, E, and NAC, its properties may influence biological and physiological dynamics in a unique manner in vivo. The protective effect of PAPLAL against ACD requires comprehensive consideration of the ability of the nanoparticles to penetrate the dermis and their subsequent bioavailability.’
- -1 The conclusion section must be added to report critical comments on the results.
Response: As suggested, we have revised the ‘Conclusions’ to include concise comments on the results.
(Lines 343-349 in Discussion) ‘In the present study, we demonstrated the important contribution of oxidative stress in ACD. Topically applicable antioxidants can be used as new therapeutic agents for ACD to replace steroids. PAPLAL has multiple protective effects against ACD, including direct antioxidant activity and the regulation of stress response genes. Since the nanoparticles contained in PAPLAL can remain stable for a long time with strong antioxidant activity, PAPLAL is expected to be applied to new ACD treatment strategies targeting oxidative stress.’
-2 A few references are from a time long past.
Response: As suggested, we have replaced some references.